# Effects of a Lifestyle Intervention in Young Women with GDM and Subsequent Diabetes

**DOI:** 10.3390/nu14245232

**Published:** 2022-12-08

**Authors:** Gang Hu, Huikun Liu, Junhong Leng, Leishen Wang, Weiqin Li, Shuang Zhang, Wei Li, Gongshu Liu, Huiguang Tian, Shengping Yang, Zhijie Yu, Xilin Yang, Jaakko Tuomilehto

**Affiliations:** 1Pennington Biomedical Research Center, 6400 Perkins Road, Baton Rouge, LA 70808, USA; 2Tianjin Women’s and Children’s Health Center, Tianjin 300071, China; 3Population Cancer Research Program, Dalhousie University, Halifax, NS B3H 4R2, Canada; 4Department of Epidemiology and Biostatistics, School of Public Health, Tianjin Medical University, Tianjin 300070, China; 5Department of Public Health, University of Helsinki, 00014 Helsinki, Finland; 6Population Health Unit, Finnish Institute for Health and Welfare, 00271 Helsinki, Finland; 7Saudi Diabetes Research Group, King Abdulaziz University, Jeddah 21589, Saudi Arabia

**Keywords:** gestational diabetes mellitus, type 2 diabetes, lifestyle intervention, postpartum

## Abstract

The purpose of this study was to examine whether a 9-month intensive lifestyle intervention could lead to weight loss and improve cardiovascular risk factors among young women with both gestational diabetes mellitus (GDM) and newly diagnosed diabetes. A total of 83 young women, who had GDM and were subsequently diagnosed as type 2 diabetes at an average of 2.6 years after delivery, participated in a 9-month intensive lifestyle intervention and a follow-up survey at 6–9 years postintervention. After the 9-month intervention, these women had a weight loss of 2.90 kg (−4.02% of initial weight), decreased waist circumference (−3.12 cm), body fat (−1.75%), diastolic blood pressure (−3.49 mmHg), fasting glucose (−0.98 mmol/L) and HbA1c (−0.72%). During the 6–9 years postintervention period, they still had lower weight (−3.71 kg; −4.62% of initial weight), decreased waist circumference (−4.56 cm) and body fat (−2.10%), but showed a slight increase in HbA1c (0.22%). The prevalence of using glucose-lowering agents increased from 2.4% at baseline to 34.6% after the 9-month lifestyle intervention, and to 48.4% at 6–9 years postintervention. A 9-month intensive lifestyle intervention can produce beneficial effects on body weight, HbA1c and other cardiovascular risk factors among young women with previous GDM who subsequently developed new diabetes.

## 1. Introduction

Gestational diabetes mellitus (GDM), defined as any degree of glucose intolerance with onset or first recognition during pregnancy, is a common pregnancy outcome [1]. GDM affects about 8% of pregnant women in the US and in China [2,3,4]. Women with prior GDM have almost a ten times higher risk of developing type 2 diabetes as compared with women with normoglycemic pregnancy, with the highest risk during 1–5 years after GDM [5]. About 20% of women with prior GDM will develop type 2 diabetes within 10 years after pregnancy, and more than 40% within 30 years [6]. Women with prior GDM also have increased risks of obesity, hypertension, dyslipidemia, metabolic syndrome, and cardiovascular disease (CVD) [7,8,9,10,11,12].

As a part of the postpartum care of GDM, American Diabetes Association (ADA) recommends a 2 h 75 g oral glucose tolerance test (OGTT) for women with recent GDM to detect persistent diabetes at 4–12 weeks postpartum and then repeatedly every 1 to 3 years thereafter [1]. Women with GDM and subsequent undiagnosed diabetes at 1–5 years postpartum are generally young and have a very low awareness of diabetes. Some studies have found that participants with newly diagnosed diabetes have a higher risk of CVD compared with those with normal glucose [13,14]. Although recent randomized controlled trials, such as the Look AHEAD (Action for Health in Diabetes) trial, have indicated that an intensive lifestyle intervention could help in losing weight and improve HbA1c among overweight or obese adults with type 2 diabetes [15], no studies have assessed short- and long-term effects of a lifestyle intervention on cardiovascular risk factors among young women with GDM and subsequent undiagnosed diabetes. The aim of this single-arm trial was to examine whether a 9-month intensive lifestyle intervention could lead to weight loss and improve cardiovascular risk factors among young women with both GDM and newly diagnosed diabetes and to investigate to what extent such effects might sustain in the long term.

## 2. Materials and Methods

### 2.1. Study Design and Participants

The present study was a single-arm 9-month trial carried out at the Tianjin Women’s and Children’s Health Center, Tianjin, China, which was part of The Tianjin Gestational Diabetes Mellitus Prevention Program (TGDMPP) [16,17,18,19]. The study design of TGDMPP, including recruitment, screening visits, and inclusion and exclusion criteria, has been described in detail elsewhere [16,17,18,19]. Briefly, 4644 women with GDM diagnosed on the basis of the 1999 World Health Organization (WHO) criteria [20] from six urban districts in Tianjin between 2005 and 2009 were recruited 1–5 years after delivery, and 1263 women (27% participation rate) aged more than 24 years old returned and completed the baseline survey administered by health workers from the study center between 2009 and 2011 [16,17,18,19]. There were no differences in the OGTT at 26–30 weeks’ gestation with regard to age (28.9 vs. 28.7 years), fasting plasma glucose (5.34 vs. 5.34 mmol/L), 2 h glucose (9.23 vs. 9.16 mmol/L), and the prevalence of impaired glucose tolerance (90.9 vs. 91.8%) and diabetes (9.1 vs. 8.2%) between the returned and unreturned women with GDM. Among the 1263 women with prior GDM who finished the baseline survey, 83 were diagnosed as having type 2 diabetes by a 2 h 75 g OGTT and confirmed by another 2 h 75 g OGTT according to the 2005 ADA criteria [21]. These 83 women with prior GDM and newly diagnosed type 2 diabetes finally participated in a 9-month intensive lifestyle intervention program and clinical visits and also completed a follow-up survey at 6–9 years postintervention (Figure 1). Ethics approval was granted by the Human Subjects Committee of the Tianjin Women’s and Children’s Health Center, and all participants provided written informed consent. This trial has been registered with ClinicalTrials.gov, number NCT01554358.

The inclusion criteria for women were: (1) age 20–49 years at the baseline survey; and (2) GDM detected between 2005 and 2009. The exclusion criteria were: (1) age < 20 or ≥50 years; (2) presence of diabetes before pregnancy; (3) taking medicines known to alter OGTT at the baseline survey; (4) presence of any chronic diseases that could seriously reduce their life expectancy or their ability to participate in the trial; (5) unable or unwilling to give informed consent or communicate with study staff; and (6) currently pregnant, or planning to become pregnant in the next two years [16].

### 2.2. Screening Visit

The screening visit included informed consent and the completion of a detailed medical eligibility questionnaire. The eligibility screening questionnaire was based on the inclusion/exclusion criteria described above. If eligible, women finished the baseline survey at the same visit. At the baseline survey, all women completed a self-administered questionnaire and underwent a physical examination. They also completed a 3-day 24 h food record using methods for dietary record collections taught by a dietitian.

### 2.3. Run-In

Before the intervention, eligible participants were given two classes in a 2-week “run-in” period on general principles of lifestyle intervention for the management of type 2 diabetes and obesity. Participants discussed their previous experiences with lifestyle changes and obstacles encountered. The accumulating evidence that showed lifestyle intervention is effective for the management of type 2 diabetes and obesity was presented. The specific lifestyle intervention program began after the run-in.

### 2.4. Lifestyle Intervention

Major elements of the intervention included five face-to-face sessions with study dietitians and two telephone calls during the 9-month lifestyle intervention program.

Dietary intervention (1–4): In the first week, each participant individually met with a dietitian who instructed the participant on how to achieve the five goals of the intervention, including (1) a reduction of 5 to 10% of initial body weight in women with body mass index (BMI) ≥ 24 kg/m^2^ by reducing at least 10% of total calories in their normal meals, and no weight loss for GDM women with BMI < 24 kg/m^2^, using the Chinese BMI classification standard [22]; (2) total fat intake < 30% of consumed energy; (3) carbohydrate intake 55–65% of consumed energy; (4) fiber intake 20–30 g per day; and (5) moderate or vigorous exercise for at least 30 min daily, seven days each week [16,17,18,19]. Based on each participant’s current status of body weight, breastfeeding, and physical activity, along with food and nutrient intakes that had been assessed at the baseline survey examination, the dietitian provided advice on how to modify their diet, which included (a) intake of appropriate energy, (b) inclusion of appropriate amounts of fish, eggs, low-fat milk, lean meat, and reduction in fatty meat and animal fat in the diet, (c) avoidance/reduction of simple sugars and refined carbohydrates, and (d) inclusion of more fiber-rich food, such as whole grains, standard grade wheat flour, corn/corn starch, brown rice, vegetables and fruits. To help participants meet these healthy diet goals, the dietitian provided the participants with a suggestive daily menu for five days. A dietitian established a minimum of five breakfasts, five lunches, and five dinners that the women were recommended to consume within the 5-day cycle menu, taking into consideration individual food preferences. Moreover, the dietitian taught each participant to select a diet that matched their specific diet prescription for macronutrients and energy and also introduced a method to exchange foods, as depicted in a handbook.

Physical activity intervention (Weeks 1–4): The physical activity goal was to gradually increase the physical activity from 15 to 30 min per day during the first four weeks. Participants had been instructed to engage in moderate or vigorous physical activity during commuting (walking or cycling to/from work) or leisure time (e.g., walking, bicycling, etc.) for at least 15 min per day, seven days per week during Week 1. The level of physical activity was planned to be increased to at least 30 min per day, seven days per week in Week 4, and maintained there during the entire trial period.

Diet and physical activity monitoring (Week 4–Month 9): Each participant completed a questionnaire on changes in major dietary and physical activity habits at the last visit, and provided their 3-day 24 h food records four times during the 9-month period for assessment by the dietitian. The dietitian reviewed questionnaires and food records, calculated the nutrient intakes using the dietary analysis software (version 2.3) developed by China CDC, provided an assessment of deviations from the suggested diets and exercises, and then offered specific suggestions at each visit. Body weight was measured at each visit.

Telephone intervention: The participants received two phone calls to encourage continued compliance with the intervention during the 9-month trial. The dietitian adjusted treatment advice to improve dietary adherence to weight loss goals. Almost 100% of mothers have mobile phone service.

### 2.5. Measurement

All study participants filled in a questionnaire at the baseline survey about their sociodemographic characteristics (age, marital status, education, income and occupation), history of GDM (values of fasting and 2 h glucose in the OGTT and treatment of GDM during the pregnancy), family history (diabetes, coronary heart disease, stroke, cancer and hypertension), medical history (hypertension, diabetes, and hypercholesterolemia), pregnancy outcomes (pre-pregnancy weight, weight gain in pregnancy, and number of children), postpartum weight and fasting glucose status, dietary habits (a self-administered food frequency questionnaire [FFQ] to measure the frequency and quantity of intake of 33 major food groups and beverages during the past year) [23], alcohol intake, smoking habits, physical activity (the frequency and duration of five domains of physical activity, including occupational, commuting [walking or cycling to/from work], leisure time [11 recreational activities], household [cooking, washing clothes, cleaning, and looking after children] and sedentary activities) [24,25], and sleep status (duration of usual sleep and night shift work).

Changes in lifestyle were measured using a questionnaire on changes in major dietary and physical activity habits at the last visit, a 3-day 24 h food record was completed at each clinical visit, and a self-administered questionnaire was completed at follow-up visits at Month 9 and Years 6–9. The performance of 3-day 24 h food records [23], the FFQ [23], and the above questionnaire on assessing physical activity [24,25] were validated in the China National Nutrition and Health Survey in 2002.

Body weight, height, waist and hip circumferences, body fat (measured using Tanita Body Composition Analyzers SC-240, Arlington Heights, IL), blood pressure, and heart rate were measured in all women using a standardized protocol at the visits of baseline, Month 9 and Years 6–9. Body weight was also measured at each clinical visit within the first 9 months. BMI was calculated as weight divided by the square of height. Peripheral venous EDTA-blood samples were collected from all participants after fasting for at least 10 h. Plasma glucose and lipid profile (total cholesterol, high-density lipoprotein [HDL] cholesterol, and triglycerides) were measured with an automatic analyzer (Toshiba TBA-120FR, Tokyo, Japan). Serum insulin was determined by an electro-chemiluminescent immunoassay (Bayer ADVIA Centaur CP, Barmen, Germany). Glycated hemoglobin (HbA1c) was measured using an Automatic Glycohaemoglobin Analyzer (ADAMS A1c HA-8160; Arkray, Kyoto, Japan).

### 2.6. Follow-Up and Outcome Indices

The primary outcomes of the trial were changes in body weight and HbA1c from baseline to Month 9 as well as from baseline to Years 6–9. The secondary outcomes were changes in waist circumference, body fat, blood pressure, fasting plasma glucose, fasting serum insulin, plasma total, HDL, and low-density lipoprotein (LDL) cholesterol, plasma triglycerides, and the use of glucose-lowering agents from baseline to Month 9 and from baseline to Years 6–9.

### 2.7. Statistical Analysis

A one-way ANOVA and chi-square test were used to analyze the differences in continuous baseline measures and in categorical measures among women with GDM and newly diagnosed diabetes who attended and absented Month 9 and Years 6–9 follow-up visits, respectively. We assessed changes in body weight, energy intake, and leisure time physical activity from baseline to Months 1, 3, 6 and 9. A two-tailed paired *t* test was used to analyze the differences in continuous measures between the two groups from baseline to Month 9 and from baseline to Years 6–9. Mean changes and their SD were calculated. An intention-to-treat analysis that included all participants was conducted in this study. A monotone missing pattern was observed among the missing values, and a multiple imputations using a regression model were performed. A *p*-value < 0.05 was considered statistically significant. Statistical analyses were performed with IBM SPSS Statistics 25.0 (IBM SPSS, Chicago, IL, USA) and SAS for Windows, version 9.4 (SAS Institute, Inc, Cary, NC, USA).

## 3. Results

A total of 68 women finished the 9-month lifestyle intervention, and 39 women attended the 6–9 years postintervention survey. Mean values of age among these participants were 32.7 at baseline, 33.8 after the 9-month lifestyle intervention, and 39.5 at the 6–9 years postintervention (Table 1). Baseline characteristics were almost similar among all participants with GDM and newly diagnosed diabetes who attended and who did not attend Month 9 and Years 6–9 follow-up visits except for a younger age group of women absenting the follow-up visits (Table 1).

Table 2 and Appendix A presented changes in body weight, energy intake and leisure time physical activity from baseline to Months 1, 3, 6 and 9. Participants who completed interventions at one of the visits at Months 1, 3, 6 and 9 lost an average of 1.82 kg weight at Month 1, 2.81 kg at Month 3, 3.21 kg at Month 6, and 2.90 kg at Month 9. Participants reduced their daily energy intake ranging from 62 up to 242 kcal and increased daily leisure time physical activity from 4.3 up to 8.5 min from baseline to Months 1–9.

During the 9-month intervention, women lost their weight of average 2.90 kg (−4.02% of initial weight), decreased their BMI (−1.12 kg/m^2^), waist circumference (−3.12 cm), body fat (−1.75%), diastolic blood pressure (−3.5 mmHg), fasting plasma glucose (−0.98 mmol/L), fasting serum insulin (−15.5 pmol/L) and HbA1c (−0.72%) (Table 3, Appendix A). During the 6–9 years postintervention, women still had lower weight (−3.71 kg; −4.62% of initial weight), decreased BMI (−1.57 kg/m^2^), waist circumference (−4.56 cm), and body fat (−2.10%), but had a slight increase in their HbA1c (0.22%). The prevalence of the use of glucose-lowering drugs increased from 2.4% at baseline to 34.6% after the 9-month lifestyle intervention, and to 48.4% at 6–9 years postintervention.

No serious adverse events occurred during the 9-month lifestyle intervention and the 6–9 years postintervention.

## 4. Discussion

Among young women with GDM and subsequent newly diagnosed diabetes at an average of 2.6 years after delivery, who took part in a 9-month intensive lifestyle intervention from baseline to follow-up, significant weight loss, decreased HbA1c, and improved cardiovascular risk factors were observed. The weight loss effect of this lifestyle intervention was sustained for at least 6 years.

Overweight and obesity are more common among patients with type 2 diabetes [26,27] as well as among women with prior GDM [16,28]. Thus, weight loss is recommended by the ADA for overweight or obese patients with type 2 diabetes [29]. The Look AHEAD trial has found that the intensive lifestyle intervention can result in weight loss and improvements of HbA1c, blood pressure, and HDL cholesterol during the intervention and a modest weight loss was maintained during the postintervention follow-up period in 5145 overweight or obese adults with type 2 diabetes, but it did not reduce cardiovascular events [15,30,31]. A recent review including 11 trials indicated that weight-loss dietary or lifestyle intervention trials led to a weight loss of < 5% of initial weight at 12 months among overweight or obese people with type 2 diabetes [32]. Only two studies had weight losses of > 5% in 12-month trials (the Mediterranean-style diet in the Esposito trial and the intensive lifestyle intervention in the Look AHEAD trial). They also had a significant decrease in HbA1c and significant improvements in lipids and blood pressure [32]. All these trials, however, focused on middle-aged or older adults with a history of type 2 diabetes [32]. No trials have assessed the short- and long-term cardiovascular effects of a lifestyle intervention among younger women with prior GDM and subsequent newly diagnosed diabetes. Nevertheless, several studies have found that people with type 2 diabetes diagnosed at < 40 years of age demonstrably increased their risk of CVD and had a reduced life expectancy [33,34]. Moreover, women with recent GDM and undiagnosed diabetes at 1–5 years postpartum are relatively young, have a very low awareness of diabetes, and have an increased risk of obesity, hypertension, dyslipidemia, metabolic syndrome, and CVD [7,8,9,10,11].

All participants in our study were relatively young, aged 32.7 years at baseline, 33.8 years after the 9-month lifestyle intervention, and 39.5 years at 6–9 years postintervention. All 83 participants had a history of GDM, and almost all of them were newly diagnosed with type 2 diabetes 2.6 years after delivery, while only 2.4% of them used glucose-lowering drugs. The present study showed that women who participated in this 9-month intensive postpartum lifestyle intervention, from baseline to follow-up, significantly lost weight, decreased HbA1c, and improved blood pressure, fasting glucose and insulin. We also found that a significant reduction in daily energy intake (227–242 kcal/d) within the first six months of intervention and an increase in leisure time physical activity (4.3–8.5 min/day) during the first nine months of intervention were associated with the significant weight loss during the 9-month intervention. The weight loss effect with a lifestyle intervention persisted for at least 6 years during the postintervention follow-up period; however, HbA1c levels slightly increased in this long-term period. Several studies have found that decreased HbA1c levels were associated with a decrease in major CVD among patients with type 2 diabetes [35,36].

An important strength of the present study is that it is the first to evaluate the short- and long-term cardiovascular effects of a lifestyle intervention among women with GDM and subsequent diabetes. This study’s participants were younger at baseline and during follow-up of the intervention, and they both had GDM and undiagnosed type 2 diabetes at an average of 2.6 years after delivery. The major limitation of the study is the single-arm trial design. Thus, we cannot distinguish the effects of a lifestyle intervention from the effects due to other reasons. Another limitation is the 82% retention rate during the 9-month intensive lifestyle intervention and the 47% retention rate at the postintervention follow-up survey at 6–9 years. However, the retention rate of the present study during the 9-month intensive lifestyle intervention was similar to that of the POUNDS Lost study (80%), another large weight loss RCT [37], and higher than those of other large GDM intervention RCTs, such as the Gestational Diabetes’ Effects on Moms study (67.3%) [38]. In addition, an intention-to-treat analysis that included all participants was conducted in the present study.

## 5. Conclusions

We found that young women with GDM and newly diagnosed diabetes who participated in a 9-month intensive lifestyle intervention had 4.0% weight loss, 0.72% decrease in HbA1c, and decreased waist circumference and blood pressure from baseline to follow-up. The weight loss effect with such an intensive lifestyle intervention persisted for at least 6 years with a weight loss of 4.6% from the initial body weight, while HbA1c slightly increased in the long term.

## Figures and Tables

**Figure 1 nutrients-14-05232-f001:**
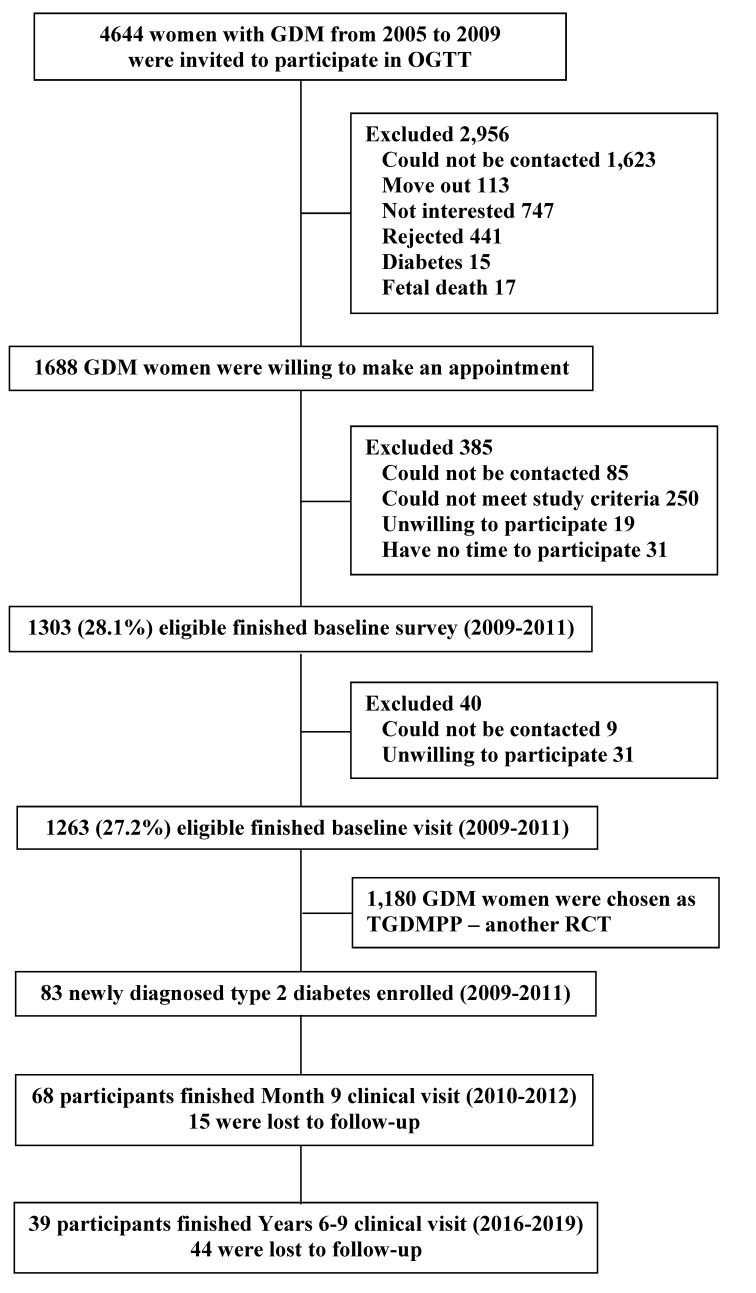
Participant flow chart.

**Table 1 nutrients-14-05232-t001:** Baseline characteristics among women with gestational diabetes mellitus (GDM) and newly diagnosed diabetes who attended and absented Month 9 and Years 6–9 follow-up visits.

	Total Samples	Month 9 visit		*p*-values	Year 6–9 Follow-Up Visit		*p*-Values
		Attend	Absence		Attend	Absence	
No. of participants	83	68	15		39	44	
Age at baseline (years)	32.7 ± 3.84	33.2 ± 3.90	30.3 ± 2.47	0.007	33.7 ± 3.83	31.7 ± 3.64	0.016
Baseline years after delivery (years)	2.63 ± 0.95	2.73 ± 0.94	2.17 ± 0.88	0.034	2.69 ± 0.95	2.58 ± 0.96	0.628
Characteristics of GDM							
Fasting plasma glucose (mmol/L)	6.01 ± 1.23	6.04 ± 1.31	5.88 ± 0.78	0.636	5.98 ± 1.41	6.04 ± 1.07	0.809
2-h plasma glucose (mmol/L)	10.5 ± 1.75	10.6 ± 1.67	10.1 ± 2.06	0.249	10.7 ± 1.84	10.3 ± 1.67	0.355
Gestational week at OGTT (weeks)	28.5 ± 2.16	28.5 ± 2.21	28.5 ± 1.83	0.971	28.3 ± 2.56	28.7 ± 1.74	0.490
Gestational age at delivery (weeks)	38.5 ± 1.80	38.4 ± 1.88	38.9 ± 1.36	0.412	38.7 ± 1.71	38.3 ± 1.88	0.286
Pre-pregnancy body mass index (kg/m^2^)	25.6 ± 3.71	25.4 ± 3.77	26.5 ± 3.36	0.276	25.3 ± 3.05	25.8 ± 4.23	0.520
Baseline survey							
Body mass index (kg/m^2^)	27.3 ± 4.04	27.1 ± 4.23	28.3 ± 2.89	0.272	27.1 ± 3.74	27.4 ± 4.32	0.738
Waist circumference (cm)	88.7 ± 9.62	88.0 ± 10.2	91.7 ± 5.97	0.170	88.3 ± 8.90	88.9 ± 10.3	0.775
Body fat (%)	37.8 ± 5.19	37.5 ± 5.46	39.5 ± 3.39	0.215	37.7 ± 4.84	38.0 ± 5.53	0.812
Systolic blood pressure (mmHg)	115 ± 15.1	115 ± 16.1	112 ± 9.57	0.405	115 ± 12.6	114 ± 17.2	0.670
Diastolic blood pressure (mmHg)	80.7 ± 11.2	81.2 ± 11.8	78.3 ± 7.96	0.360	81.6 ± 9.96	79.9 ± 12.3	0.484
Fasting plasma glucose (mmol/L)	7.64 ± 2.17	7.73 ± 2.27	7.23 ± 1.61	0.425	7.58 ± 1.85	7.69 ± 2.43	0.825
2-h plasma glucose (mmol/L)	14.1 ± 3.01	14.3 ± 3.15	13.3 ± 2.14	0.250	14.3 ± 2.99	14.0 ± 3.06	0.702
Fasting serum insulin (pmol/L)	82.6 ± 68.0	82.9 ± 72.9	82.4 ± 45.4	0.985	88.2 ± 87.5	77.9 ± 46.0	0.492
HbA1c (%)	6.91 ± 1.23	6.94 ± 1.29	6.75 ± 0.92	0.577	6.83 ± 1.16	6.98 ± 1.30	0.591
Plasma total cholesterol (mmol/L)	4.92 ± 0.88	4.91 ± 0.91	4.95 ± 0.75	0.862	4.74 ± 0.90	5.07 ± 0.84	0.078
High-density lipoprotein cholesterol (mmol/L)	1.26 ± 0.27	1.28 ± 0.25	1.18 ± 0.32	0.171	1.28 ± 0.25	1.25 ± 0.28	0.259
Low-density lipoprotein cholesterol (mmol/L)	2.78 ± 1.02	2.85 ± 0.79	2.47 ± 1.72	0.189	2.73 ± 0.73	2.83 ± 1.23	0.692
Plasma triglycerides (mmol/L)	1.90 ± 2.31	1.69 ± 0.78	2.86 ± 5.20	0.077	1.60 ± 0.78	2.17 ± 3.08	0.656
Use of glucose-lowering agents (%)	2.4	2.9	0	0.527	2.6	2.3	0.595
Leisure time physical activity (min/day)	2.81 ± 7.96	3.31 ± 8.68	0.59 ± 2.04	0.234	2.63 ± 8.40	2.99 ± 7.64	0.839
Dietary intake ^a^							
Energy intake (kcal/day)	1708 ± 418	1720 ± 420	1653 ± 420	0.574	1738 ± 402	1681 ± 436	0.536
Protein (% energy)	16.0 ± 2.36	16.2 ± 2.37	15.3 ± 2.23	0.191	16.0 ± 2.36	16.0 ± 2.29	0.956
Fat (% energy)	33.3 ± 6.44	33.9 ± 6.21	30.5 ± 6.95	0.063	33.8 ± 6.85	32.9 ± 6.10	0.523
Carbohydrate (% energy)	50.7 ± 7.12	49.9 ± 6.75	54.2 ± 7.91	0.034	50.2 ± 7.82	51.1 ± 6.49	0.576

Data are presented as mean ± SD unless otherwise indicated. A one-way ANOVA and chi-square test were used to analyze the differences in continuous baseline measures and in categorical measures. ^a^ Dietary intake assessed was by 3-day 24 h food records.

**Table 2 nutrients-14-05232-t002:** Changes in body weight, energy intake, leisure time, and physical activity from baseline to the end of Months 1, 3, 6 and 9.

	Month 1	Month 3	Month 6	Month 9
No. of participants	83	83	83	83
Change in weight				
(kg)	−1.82 ± 2.51	−2.81 ± 3.19	−3.21 ± 3.73	−2.90 ± 3.77
Percent reduction in initial weight	−2.54 ± 3.62	−3.97 ± 4.61	−4.52 ± 5.37	−4.02 ± 5.34
Change in energy intakes (kcal/day) ^a^	−242 ± 451	−240 ± 414	−227 ± 416	−61.7 ± 371
Change in leisure time physical activity (min/day)	4.27 ± 16.8	4.97 ± 12.5	7.61 ± 20.4	8.46 ± 24.3

Data are presented as changes in mean ± SD unless otherwise indicated. A monotone missing pattern was observed among the missing values, and multiple imputations using a regression model were performed. ^a^ Dietary intake was assessed by 3-day 24 h food records.

**Table 3 nutrients-14-05232-t003:** Changes in selected clinical and metabolic variables from baseline to the end of Month 9 and Years 6–9 visits.

	Baseline	Month 9	Changes from Baseline to Month 9	*p*-Values	Years 6–9	Changes from Baseline to Years 6–9	*p*-Values
No. of participants	83	83	83		83	83	
Age at baseline or during follow-up (years)	32.7 ± 3.84	33.8 ± 3.94			39.5 ± 3.98		
Follow-up from baseline (years)		1.12 ± 0.34			6.79 ± 1.13		
Body weight (kg)	70.2 ± 11.1	67.3 ± 11.0	−2.90 ± 3.77	< 0.001	66.5 ± 9.03	−3.71 ± 6.02	< 0.001
Percent change in initial weight			−4.02 ± 5.34			−4.62 ± 8.30	
Body mass index (kg/m^2^)	27.3 ± 4.04	26.2 ± 4.04	−1.12 ± 1.48	< 0.001	25.7 ± 3.13	−1.57 ± 2.35	< 0.001
Waist circumference (cm)	88.7 ± 9.62	85.6 ± 9.84	−3.12 ± 5.94	< 0.001	84.1 ± 8.48	−4.56 ± 7.41	< 0.001
Body fat (%)	37.9 ± 5.19	36.1 ± 5.57	−1.75 ± 2.28	< 0.001	35.8 ± 4.51	−2.10 ± 3.20	< 0.001
Systolic blood pressure (mmHg)	115 ± 15.1	116 ± 14.8	1.63 ± 12.7	0.305	119 ± 16.4	3.88 ± 14.4	0.072
Diastolic blood pressure (mmHg)	80.7 ± 11.2	77.2 ± 11.0	−3.49 ± 9.93	0.009	79.0 ± 10.7	−1.64 ± 8.22	0.155
Fasting plasma glucose (mmol/L)	7.64 ± 2.17	6.66 ± 1.99	−0.98 ± 1.80	< 0.001	8.28 ± 2.72	0.64 ± 2.81	0.099
Fasting serum insulin (pmol/L)	82.6 ± 68.0	67.0 ± 48.8	−15.5 ± 58.3	0.031	71.8 ± 41.0	−10.8 ± 49.3	0.073
HbA1c (%)	6.91 ± 1.23	6.19 ± 1.21	−0.72 ± 1.02	< 0.001	7.13 ± 1.54	0.22 ± 1.76	0.345
Total cholesterol (mmol/L)	4.92 ± 0.88	4.86 ± 0.94	−0.06 ± 0.69	0.507	5.12 ± 1.32	0.20 ± 1.12	0.241
High-density lipoprotein cholesterol (mmol/L)	1.26 ± 0.27	1.24 ± 0.30	−0.02 ± 0.24	0.470	1.31 ± 0.29	0.05 ± 0.25	0.135
Low-density lipoprotein cholesterol (mmol/L)	2.78 ± 1.02	2.78 ± 1.04	0.00 ± 0.78	0.968	2.97 ± 0.85	0.19 ± 0.97	0.136
Triglycerides (mmol/L)	1.90 ± 2.29	1.74 ± 2.54	−0.16 ± 1.85	0.508	1.56 ± 1.74	−0.34 ± 2.41	0.221
Use of glucose-lowering agents (%)	2.41	34.6	32.2		48.4	46.0	

Data are presented as mean ± SD unless otherwise indicated. A monotone missing pattern was observed among the missing values, and multiple imputations using a regression model were performed. A two-tailed paired *t* test was used to analyze the differences in continuous measures between the two groups.

## Data Availability

The datasets used and/or analyzed during the current study are available from the corresponding author on reasonable request.

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
