# Peer review of "Effects of a Lifestyle Intervention in Young Women with GDM and Subsequent Diabetes"

_nutrients, 2022, doi:10.3390/nu14245232_

Round 1
Reviewer 1 Report
This paper titled “Effects of a lifestyle intervention in young women with GDM and subsequent diabetes” is interesting. This manuscript need major revising.
My comments are as follows:
- The references are old, authors should focus on the articles published in recent years.
- The discussion section should be improved. Authors should compare current data with previous data.
- English should be improved.
Author Response
Response to Reviewer 1’s Comments
Point 1: The references are old, authors should focus on the articles published in recent years.
Response 1: Thanks for your comments. We have updated some references of recent years.
Point 2: The discussion section should be improved. Authors should compare current data with previous data.
Response 2: We have modified the Discussion section based on your suggestion.
Point 3: English should be improved.
Response 3: We have checked the mansucript by one native English speaker.

Reviewer 2 Report
I would suggest adding calendar years to Figure 1 or a sentence in the body of the manuscript to reflect that the study did not end until 2018(?) as the initial 2005-2009 timeframe leaves the reader with an unclear perspective of the timeliness and relevance of the work. Otherwise interesting result. Very striking participation at 6-9 years. Seems low.
Author Response
Response to Reviewer 2’s Comments
Point 1: I would suggest adding calendar years to Figure 1 or a sentence in the body of the manuscript to reflect that the study did not end until 2018(?) as the initial 2005-2009 timeframe leaves the reader with an unclear perspective of the timeliness and relevance of the work.
Response 1: Thanks for your comments. We have added calendar years in Figure 1.
Point 2: Otherwise interesting result. Very striking participation at 6-9 years. Seems low.
Response 2: Thanks for your positive comments. We have added more details about this low retention rate at 6-9 years in the Limitation section.

Reviewer 3 Report
The aim of this single-arm trial was to examine whether a 9-month intensive lifestyle intervention 56 can lead to weight loss and improve cardiovascular risk factors among young women with both GDM and newly diagnosed diabetes, and to investigate to what extent such 58 effects may sustain in a long term. The found that young women with GDM and newly diagnosed diabetes who particpated in a 9-month intensive lifestyle intervention led to a 4.0% weight loss, a 0.72% de-301 crease in HbA1c, and also decreases in waist circumference and blood pressure from baseline to follow-up. The weight loss effect with such an intensive lifestyle intervention seems to persist for at least 6 years with weight loss as 4.6% of the initial body weight, while HbA1c may slightly increase in the long term.
The introduction is well written , with adequate bibliographic references and stating the hypothesis of the study
The methodology is very exhaustive, widely described, which would allow the study to be carried out by another research group. The absence of a control group that performs usual care would be necessary to increase the validity of the results. This fact is an important limitation. Results are clearly described and easy to understand The discussion is correct, adapting to the results obtained.
Author Response
Response to Reviewer 3’s Comments
Point 1: The aim of this single-arm trial was to examine whether a 9-month intensive lifestyle intervention can lead to weight loss and improve cardiovascular risk factors among young women with both GDM and newly diagnosed diabetes, and to investigate to what extent such effects may sustain in a long term. The found that young women with GDM and newly diagnosed diabetes who particpated in a 9-month intensive lifestyle intervention led to a 4.0% weight loss, a 0.72% decrease in HbA1c, and also decreases in waist circumference and blood pressure from baseline to follow-up. The weight loss effect with such an intensive lifestyle intervention seems to persist for at least 6 years with weight loss as 4.6% of the initial body weight, while HbA1c may slightly increase in the long term.
The introduction is well written , with adequate bibliographic references and stating the hypothesis of the study
The methodology is very exhaustive, widely described, which would allow the study to be carried out by another research group.
Response 1: Thanks for your positive comments.
Point 2: The absence of a control group that performs usual care would be necessary to increase the validity of the results. This fact is an important limitation.
Response 2: Thanks for your comments. We have added more details about this single-arm trial design in the Limitation section.
Point 3: Results are clearly described and easy to understand. The discussion is correct, adapting to the results obtained.
Response 3: Thanks for your positive comments.

Round 2
Reviewer 1 Report
accept